Validation of CDC45 as a novel biomarker for diagnosis and prognosis of gastric cancer

Wu Lihua 1 wulihua0812@163.com
Gao Gan 2 3
Mi Hui 4
Luo Zhou 1
Wang Zheng 1
Liu Yongdong 1
Wu Liangyan 1
Long Haihua 1
Shen Yongqi 1 gxnnsyq@163.com
1 Affiliated Liutie Central Hospital of Guangxi Medical University , Liuzhou , China
2 Clinical Laboratory, Liuzhou Hospital of Guangzhou Women and Children’s Medical Center , Liuzhou, Guangxi , China
3 Guangxi Clinical Research Center for Obstetrics and Gynecology , liuzhou, Guangxi , China
4 Changzhi People’s Hospital , Changzhi , china
Amdare Nitin
Electronic publication date: 2024 Mar 18
Publication date: 2024
Volume: 12
Electronic Location ID: e17130
Received 2023 Oct 25; Accepted 2024 Feb 27
Copyright: © 2024 Wu et al.
Copyright year: 2024
Copyright holder: Wu et al.
License: This is an open access article distributed under the terms of the Creative Commons Attribution License, which permits unrestricted use, distribution, reproduction and adaptation in any medium and for any purpose provided that it is properly attributed. For attribution, the original author(s), title, publication source (PeerJ) and either DOI or URL of the article must be cited.
License URL: https://creativecommons.org/licenses/by/4.0/

Keywords: Gastric cancer, CDC45, Biomarker, Diagnostic, Biomarker

Funding: Guangxi Zhuang Autonomous Region Health and Family Planning Commission Z20210651 and Z20210712 This study was supported by the Guangxi Zhuang Autonomous Region Health and Family Planning Commission: Z20210651; Z20210712. The funders had no role in study design, data collection and analysis, decision to publish, or preparation of the manuscript.

==============================
Background

Cell division cycle protein 45 (CDC45) has been demonstrated to play vital roles in the progression of various malignancies. However, the clinical significance of CDC45 in gastric cancer (GC) remains unreported.

Method

In this study, we employed the TCGA database and the TCGA & GTEx dataset to compare the mRNA expression levels of CDC45 between gastric cancer tissues and adjacent or normal tissues (p < 0.05 was considered statistically significant), which was further validated in multiple datasets including GSE13911, GSE29272, GSE118916, GSE66229, as well as RT-qPCR. Furthermore, we harnessed the Human Protein Atlas (HPA) to evaluate the protein expression of CDC45, which was subsequently verified through immunohistochemistry (IHC). To ascertain the diagnostic utility of CDC45, receiver operating characteristic (ROC) curves and the area under the ROC curve (AUC) were calculated in TCGA database, and further validated it in TCGA & GTEx and GSE66229 datasets. The Kaplan–Meier method was used to reveal the prognostic importance of CDC45 in The Cancer Genome Atlas (TCGA) database and authenticated through the GSE66229, GSE84433, and GSE84437 datasets. Through cBioPortal, we identified co-expressed genes of CDC45, and pursued enrichment analysis. Additionally, we availed gene set enrichment analysis (GSEA) to annotate the biological functions of CDC45.

Results

Differential expression analysis revealed that CDC45 was significantly upregulated at both the mRNA and protein levels in GC (all p < 0.05). Remarkably, CDC45 emerged as a promising prognostic indicator and a novel diagnostic biomarker for GC. In a comprehensive the drug susceptibility analysis, we found that patients with high expression of CDC45 had high sensitivity to various chemotherapeutic agents, among which 5-fluorouracil, docetaxel, cisplatin, and elesclomol were most evident. Furthermore, our findings suggested a plausible association between CDC45 and immune cell infiltration. Enrichment analysis revealed that CDC45 and its associated genes may play crucial roles in muscle biofunction, whereas GSEA demonstrated significant enrichment of gene sets pertaining to G protein-coupled receptor ligand binding and G alpha (i) signaling events.

Conclusion

Our study elucidates that upregulation of CDC45 is intricately associated with immune cell infiltration and holds promising potential as a favorable prognostic marker and a novel diagnostic biomarker for GC.

Introduction

Gastric cancer (GC) emerges as a prevailing malignancy in the gastrointestinal tract, posing notable morbidity and mortality rates (Sung et al., 2021). Gastric cancer accounts for 10% of global cancer-related fatalities in 2018, positioning it as the sixth leading cause of death. Research indicates that early gastric cancer (EGC) exhibits a more favorable prognosis (Wan et al., 2018). Specifically, the survival rate for postoperative EGC stands at 90% after a 5-year period. However, the absence of symptoms in patients with EGC contributes to a remarkably low rate of early detection. Nevertheless, conventional diagnostic approaches for gastric cancer possess inherent limitations (Smyth & Moehler, 2019). Consequently, the establishment of a precise and timely diagnosis for gastric cancer holds considerable importance.

The conventional diagnostic methods for gastric cancer possess inherent limitations. Frequently employed imaging techniques, including X-ray barium meal examination, gastroscopy, and ultrasound offer partial diagnostic information, but their sensitivity and specificity are constrained, thereby increasing the likelihood of missed or erroneous diagnoses. Certain serum markers, such as AFP and CEA, are utilized for gastric cancer screening and monitoring, yet their specificity is suboptimal and susceptible to external factors, resulting in an elevated misdiagnosis rate. In addition, traditional chemotherapy of GC remains to be the mainstay treatment, albeit with limited efficacy (Ajani et al., 2016). Hence, the quest for an innovative identifier and therapeutic indicator holds immense importance in the pursuit of early diagnosis and enhanced treatment approaches for gastric cancer.

Acting as an important component of the eukaryotic DNA15 helicase, cell division cycle protein 45 (CDC45) has a major influence on DNA replication by binding to the replication protein A (RPA) and guiding RPA to bind to single-stranded DNA (Szambowska et al., 2017). Recent studies have confirmed that CDC45 was abnormally expressed in various of tumors and closely related to their occurrence, development, and prognosis (Huang et al., 2019; Fu et al., 2022; He et al., 2021; Hu et al., 2019). Silence of CDC45 leads to inhibition of thyroid papillary carcinoma cell proliferation by G1 arrest and induction of apoptosis (Sun et al., 2017). Zhang, Liu & Zhang (2021) discovered overexpression of CDC45 suppresses the PI3K/AKT pathway to inhibit cell proliferation in acute leukemia. Highly expressed CDC45 was found to correlate with poorer prognosis and overall survival in hepatocellular carcinoma (Lu et al., 2021). However, there are few studies on the role of CDC45 in gastric cancer. Our study comprehensively analyzed the expression, diagnostic and prognostic value of CDC45, and further explored its related immune infiltration and function enrichment pathways. The purpose of this study is to uncover the CDC45 function and its clinical application value in GC.

Materials and Methods

Expression and transcription analysis

The TIMER2.0 (Li et al., 2017) was utilized to investigate the CDC45 expression among 33 different cancer types in The Cancer Genome Atlas (TCGA) project. Clinical and RNA-seq data of 375 gastric cancer patients and 32 normal samples were obtained from the TCGA database through the UCSC Xena website (https://xena.ucsc.edu/), and 210 normal samples were obtained from GTEX databases (https://www.gtexportal.org/home/). Additionally, four sets of gastric cancer chip datasets, namely GSE13911, GSE29272, GSE118916, and GSE66229, were downloaded from the GEO (https://www.ncbi.nlm.nih.gov/geo/) database and utilized as validation sets to investigate the differential expression of CDC45 in this study. The log2 [TPM (transcripts per million) +1] or log2 [FPKM (fragments per kilo base per million) +1] transformed expression data were employed for data analysis. The analysis process of this study is presented in Fig. 1. Detailed information about the data sources can be found in Table S1.

Figure 1 Flow chart of the study.

Diagnostic and survival analysis

The diagnostic value of CDC45 was evaluated employing the pROC package (Robin et al., 2011) in R version 3.6.3 (R Core Team, 2020). The resulting ROC curves were plotted using the ggplot (Wickham, 2009). To verify the diagnostic performance, the GSE66229 dataset, which included 300 gastric cancer samples and 100 paired normal gastric tissue samples, was employed. The correlation between CDC45 gene expression and overall survival was analyzed using the Kaplan-Meier Plotter in the R package. Furthermore, three additional GC chip datasets (GSE66229, GSE84433, and GSE84437) were obtained from the GEO database and used for prognostic validation.

Clinical specimen collection and RT-PCR analysis

Human GC tissue specimens and adjacent normal mucosa tissues were obtained from GC patients who underwent surgical treatment and were confirmed by pathology in Liutie Central Hospital (Liuzhou, China). Samples that had undergone radiotherapy or chemotherapy as well as those with unclear pathological diagnosis and incomplete clinical data were excluded. The clinicopathological characteristics are shown in Table S2. Tissues were stored in RNAlater solution (Qiagen, Valencia, CA, USA) at −80 °C. All procedures were approved by the Institutional Review Board of Liuzhou Liutie Central Hospital (KY2023-075-01). All patients provided written informed consent. The study methods conformed to the standards established by the Declaration of Helsinki.

In preparation for RNA extraction, frozen tissues (n = 12) were then crushed in liquid nitrogen. The total RNA from these tissues (100 mg per microdissection sample) was extracted using the SteadyPure Quick RNA Extraction Kit (AG21023; Accurate Biology, Shenzhen, China) following the instructions provided by the manufacturer. DNase I (AG12001; Accurate Biology, Shenzhen, China) was applied to digest DNA contaminants. The concentrations of total RNA were quantified using a NanoQ micro-volume Spectrophotometer (CapitalBio, Beijing, China). The reverse transcription of mRNA (1 μg RNA per samples) was carried out using the GoScript™ Reverse Transcription System (A5001; Promega, Madison, WI, USA). The mRNA expression levels were measured using the GoTaq® Master Mix kit (A6001; Promega) on a Roche LightCycler 480 II Real-time fluorescence quantitative PCR machine. The primers of RT-qPCR were designed in the website Primer Bank (https://pga.mgh.harvard.edu/primerbank/index.html). Primers were synthesized by Shangya Biotechnology (Fuzhou, China) and GAPDH was used as a reference following the 2−ΔΔCT method. CDC45: forward: 5′-AGAGCATAAAGAACAGTTCCGCTA-3′; reverse, 5′-GACTGGCCTGTGAGTGTCAC-3′.

Immunohistochemistry (IHC)

Immunohistochemical (IHC) staining of gastric tissues was carried out on 3-µm-thick sections of formalin-fixed and paraffin-embedded (FFPE) tissue. The staining procedure followed standard protocols and utilized the EliVision™ Plus kit (Maixin Biotechnology, Fuzhou, China) as described previously (Chen et al., 2023). The sections were stained using antibodies against CDC45 (JJ091-04) (1:200) (Hangzhou HuaAn, Biotechnology, China) according to the manufacturers’ instructions. Hematoxylin was used for counterstaining in IHC. IHC staining were observed with a Leica light microscope (Leica DMI4000 B; Leica, Wetzlar, Germany).

Somatic mutations and drug sensitivity analysis

Somatic mutation data of the TCGA-STAD cohort (n = 396) was acquired from TCGA GDC Data Portal (https://portal.gdc.cancer.gov/). Analysis and visualization of the differences in markers of TCGA molecular classification between the high and low CDC45 expression groups were conducted by the “maftools” package (Yang et al., 2023). The cancer-related chemotherapeutic drug sensitivity was predicted via the Genomics of Drug Sensitivity Database following the previous study (Tang et al., 2020).

Immune infiltration analysis

For the analysis of the relative abundance of 22 immune cell subtypes among TCGA-STAD, the CIBERSORTx website (https://cibersortx.stanford.edu/) was used. Additionally, the relationship between the expression of CDC45 and immune infiltration in gastric cancer (GC) was investigated (Liu et al., 2018). Based on ESTIMATE, a ratio of immune-stromal components was calculated in each sample of cancer, and three scores were displayed: ImmuneScore, StromalScore, and ESTIMATEScore.

Co-expressed analysis

To identify the 255 co-expressed genes of CDC45 (Spearman’s correlation value > 0.3; p < 0.05), the “Co-expression” module of cBioPortal (https://www.cbioportal.org/) was utilized (Ding et al., 2022). From this set, the top six co-expressed genes were selected based on their cor-values. Detailed information about the data sources of co-expressed genes can be found in Table S3. Subsequently, the correlation analysis of these genes was visualized in cBioPortal. Additionally, the survival analysis of the top six co-expressed genes was conducted in GEPIA2 (http://gepia2.cancer-pku.cn/#index).

Pathway and enrichment analysis

To elucidate the potential functions of CDC45, DEGs (differentially expressed genes) between the CDC45 high and CDC45 low groups were identified using the Limma package, with the criteria of an adjusted p-value < 0.05 and a |logFC| > 1.5. Additionally, the “Cluster Profiler” R package was employed to conduct and visualize KEGG and GO (Gene Ontology) analysis, including molecular function (MF), biological process (BP), and cell composition (CC). Enrichment to a meaningful pathway was determined based on an adjusted p-value < 0.05 (Yu et al., 2012). Furthermore, Gene Set Enrichment Analysis (GSEA) (https://www.gsea-msigdb.org) was performed to identify functional and biological pathways associated with low and high expression of CDC45, using the DEGs sets between the CDC45 high and CDC45 low groups (Subramanian et al., 2005). The gene sets for pathway and enrichment analysis were listed in Table S4.

Statistical analysis

The statistical analysis was conducted using R software (version 4.0.3 & 3.6.3; R Core Team, 2020). The comparison of differential expression between cancer tissues and adjacent tissues was performed using a paired t-test. The two groups were compared using the Mann–Whitney U test. Correlation analyses were conducted using Spearman’s correlation. A significance level of p < 0.05 was used to determine statistical significance.

Results

Upregulated expression of CDC45 in gastric cancer

Initially, the expression of CDC45 was investigated across various cancer types using RNA-seq data from TCGA. It was observed that CDC45 expression was upregulated in different malignancies (Fig. 2A), particularly in gastric cancer (GC) where mRNA levels were significantly higher compared to both normal tissues and adjacent normal tissues (Figs. 2B–2D). To further validate these findings, CDC45 expression was assessed in multiple independent datasets (GSE13911, GSE29272, GSE118916, and GSE66229), consistently demonstrating significantly elevated levels of CDC45 expression in GC tissues (Figs. 2E–2H). In addition to confirming the differential expression of CDC45via qRT-PCR, as depicted in Fig. 2J. We further investigated the protein expression levels of CDC45. Our analysis, utilizing data from the Human Protein Atlas (HPA) (Fig. 2I) and validated through immunohistochemistry (IHC) analysis as shown in Fig. 2K, revealed significant disparities between normal and GC tissues. Collectively, these findings provide compelling evidence that patients with GC exhibit heightened CDC45 expression at both the mRNA and protein levels, thereby implying its potential involvement in the pathogenesis of GC.

Figure 2 Expression of CDC45 in gastric cancer.

(A) CDC45 mRNA expression level in pan-cancers analyzed in TCGA database. (B and C) The transcription levels of CDC45 in TCGA-STAD (GC vs NC) (B) and in matched GC samples from TCGA (C). (D) Differential expression of CDC45 between GC and NC (TCGA & GTEx). (E–G) Differential expression of CDC45 in tumor and adjacent normal tissues was verified in GSE13911 (E), GSE29272 (F), and GSE118916(G) datasets. (H) Differential expression of CDC45 in tumor and normal tissues was verified in the GSE66229 dataset. (I) The CDC45 protein expression level in (HPA) database. (J) Differential expression of CDC45 was validated in adjacent normal tissues and GC samples by RT-qPCR. (K) The differential expression of CDC45 protein was verified by immunohistochemistry. (***p < 0.001, **p < 0.01, *p < 0.05).

Potential significance of CDC45 in GC diagnosis and prognosis

Based on the aforementioned findings, it can be inferred that the overexpression of CDC45 may possess significant diagnostic and prognostic implications in gastric cancer. In order to substantiate this hypothesis, a comprehensive analysis and validation were conducted using multiple datasets. Notably, the diagnostic potential of CDC45 was conducted rigorously evaluated in the TCGA cohort, wherein it exhibited a strong diagnostic capability to discriminate gastric cancer (GC) cases. This was evidenced by the receiver operating characteristic (ROC) analysis, which yielded an impressive area under the curve (AUC) value of 0.911 (95% CI [0.873–0.948]) (Fig. 3A). Moreover, within the training set (TCGA+GTEx), encompassing both cancerous and non-cancerous samples, CDC45 demonstrated exceptional diagnostic accuracy. The ROC analysis further substantiated the reliability of CDC45 as a diagnostic marker for gastric cancer (GC), as evidenced by a robust area under the curve (AUC) exceeding 0.950 (Fig. 2B). To reinforce our findings, we conducted an independent validation in the GSE66229 cohort. Once again, CDC45 exhibited a substantial level of efficacy in distinguishing between GC and normal samples, with an AUC of 0.928 (95% CI [0.903–0.954]) (Fig. 3C). AFP and CEA are commonly used tumor markers in gastrointestinal tumors. Based on the TCGA-STAD data, we plotted the ROC curves of AFP and CEA for the diagnosis of gastric cancer (Fig. S1). The results showed that the AUC area of CEA and AFP in the diagnosis of gastric cancer was 0.669, 0.593, respectively, which was much lower than CDC45 (AUC = 0.911). Additionally, the effectiveness of CDC45 in differentiating between early tumor pathology (stage I and II) and normal controls was confirmed through ROC analysis (Fig. 3D). Collectively, our findings consistently indicate the up-regulation of CDC45 in GC and emphasize its potential as a valuable diagnostic biomarker.

Figure 3 Potential diagnostic and prognostic value of CDC45.

(A and B) ROC curves indicated the underlying diagnostic efficiency of CDC45 for GC based on TCGA (A) and TCGA & GTEx (B) databases. (C) The diagnostic performance of CDC45 for gastric cancer was validated in GSE66229. (D) The value of CDC45 in early diagnosis of gastric cancer was analyzed in TCGA database. (E) Relationship between CDC45 mRNA expression and prognosis in the TCGA cohort. (F–H) The prognostic value of CDC45 was validated in GSE66229(F), GSE84433 (G) and GSE84437 (H) datasets.

To assess the prognostic value of CDC45 in patients with gastric cancer (GC), Kaplan-Meier (KM) survival curves were constructed using the TCGA, GSE66229, GSE84433, and GSE84437 datasets. In the TCGA-STAD dataset (Fig. 3E), patients with high expression of CDC45 exhibited a significantly lower risk of mortality (p = 0.014; HR = 0.66) compared to those with low expression. This trend was also observed in the GSE66229 cohort (Fig. 3F), where high CDC45 expression was associated with a better OS outcome (p = 0.028; HR = 0.70). To ensure the robustness of our findings, we further validated our results in two additional independent datasets. The validation of the GSE84433 dataset consistently supported our initial findings, demonstrating that GC patients in the CDC45 high-expression group had a favorable overall survival (p = 0.015; HR = 0.68) (Fig. 3G). Likewise, the GSE84437 dataset also confirmed the association between high CDC45 expression and improved overall survival in GC patients (p = 0.025; HR = 0.73) (Fig. 3H). Taken together, these findings highlight the potential clinical significance of abnormal CDC45 expression as a prognostic indicator for patients with GC.

Somatic mutations landscapes analyses

We further conducted a comprehensive examination of the genomic characteristics of CDC45-based GC subgroups (Fig. 4A). Our analysis revealed that the high-CDC45 group exhibited significant mutations in the following five genes: TTN, MUC16, SYNE1, ARID1A, and CSMD3. This finding suggests a potential association between elevated CDC45 mRNA expression and the genomic stability of gastric cancer (GC) patients. Previous research has indicated variations in the efficacy of chemotherapy among GC patients based on their molecular classifications (Sohn et al., 2017). To further explore this phenomenon, we assessed the sensitivity of patients in the CDC45 high and low expression groups tot chemotherapy. The results demonstrated that patients in the CDC45 high expression group exhibited higher sensitivity to chemotherapeutic drugs, including 5-Fluorouracil, docetaxel, cisplatin, and elesclomol (Figs. 4B–4E). This suggests that docetaxel, paclitaxel, and cisplatin may be appropriate chemotherapeutic regimens for GC patients with high CDC45 expression.

Figure 4 Relationship between CDC45 expression and somatic mutations and sensitivity to chemotherapy drugs of GC patients.

(A) Association between CDC45 mRNA expression and somatic mutations of GC. (B) Relationship between CDC45 expression and chemo sensitivity in the TCGA-STAD cohort. (**p < 0.01, ****p < 0.0001).

Correlations between CDC45 expression and immune infiltration in GC

In this study, we employed the CIBERSORT method to examine the association between CDC45 expression and tumor-infiltrating immune cells in gastric cancer (GC). The 375 GC samples were categorized into high and low expression groups based on the median value of CDC45 expression, and the relative abundance of 22 immune cell types was analyzed. The distribution and correlations of these immune cell types in GC are depicted in Figs. 5A and 5B. The findings of the study indicate that there is a significant association between high CDC45 expression and increased infiltration of various immune cell types, including M0 macrophages, M1 macrophages, NK resting cells, activated CD4+ memory T cells, and T follicular helper cells (Fig. 5B, all p < 0.05). Conversely, low CDC45 expression is associated with other immune cell types, such as naive B cells, memory B cells, resting CD4+ memory T cells, regulatory T cells, plasma cells, activated NK cells, monocytes, and resting mast cells (Fig. 5B, all p < 0.05). Additionally, the results demonstrate a positive correlation between CDC45 expression and the infiltration of antitumor cells, including M0 macrophages, M1 macrophages, and activated CD4+ memory T cells (Fig. 5C, all p < 0.05), while low CDC45 expression was associated with increased infiltration of immunosuppressive cells, particularly regulatory T cells (Fig. 5C, all p < 0.05). These findings suggest that GC patients with low CDC45 expression may have a stronger ability to evade immune responses.

Figure 5 The relationship between the CDC45 expression and tumor-infiltration immune cells.

(A) Bar plot showed the relative content of 22 immune cells in GC samples. (B) Infiltration of 22 immune infiltrating cells between GC patients with low and high CDC45 expression obtained by CIBERSORTx. (C) Scatterplot showed the correlation between CDC45 and immune cells. (D–F) Correlations between CDC45 expression and StromalScore (D), ImmuneScore (E), and ESTIMATEScore (F). (*p < 0.05, **p < 0.01, ***p < 0.001).

The ESTIMATE method was employed to calculate immune scores, stromal scores, and ESTIMATE scores for each sample. Notably, significant negative correlations were observed between CDC45 expression and stromal score, immune score, and ESTIMATE score in GC (Figs. 5D–5F), indicating that CDC45 exerts a substantial influence on the infiltration levels of stromal and immune cells in GC.

Genes co-expressed with CDC45 in GC

To identify the top co-expressed genes with CDC45, the CBioPortal web server was utilized to analyze three distinct studies from TCGA: TCGA Firehose Legacy, TCGA Nature 2014, and TCGA Pan-Cancer Atlas. Through the examination of these studies, a total of 255 genes were identified as exhibiting positive co-expression with CDC45 representing the overlapping genes found within the three databases (Fig. 6A). The Spearman’s correlation values for this co-expression were presented in Fig. 6B. To ascertain the most highly co-expressed genes with CDC45, a ranking was established based on adjusted p values. Consequently, the top six co-expressed genes were determined through this analysis. It is plausible that these genes possess a robust association with CDC45 expression and potentially fulfill significant functions within shared biological processes or pathways. The results of the correlation analysis demonstrated a significant positive correlation between CDC45 and several genes, including protein disulfide-isomerase A5 (CDCA5), denticle less E3 ubiquitin protein ligase homolog (DTL), Non-SMC condensing I complex subunit H (NCAPH), RAN binding protein 1 (RANBP1), Origin recognition complex subunit 1 (ORC1), and lRAD54L (Fig. 6C). These findings suggest that the expression levels of CDC45 and these genes are likely to be regulated in a coordinated manner, indicating potential functional relationships or shared regulatory mechanisms. Survival analysis indicated higher expression of CDCA5, NCAPH and RAD54L were significantly correlated with more favorable prognosis (p = 0.022, 0.037 and 0.042, respectively).

Figure 6 Co-expression analyses for the key genes.

(A) 255 co-expressed genes were screened out as the mutual part in Firehose legacy, Nature 2014 and Pan Cancer Atlas. (B) Coexpressed genes with CDC45 were illustrated via the heat map in GC. (C) Spearman correlation analysis of the top six co-expressed genes positively correlated with CDC45 expression. (D) Kaplan-Meier plot analyzed overall survival in differential expression of co-expressed genes in GC.

Function enrichment analysis of potential CDC45 regulatory network in GC

A total of 56,494 genes were examined (Table S4), where in differential expression genes were visualized using the volcano plot (Fig. 7A). In order to gain further insights into the role of CDC45 and its potential interactions with other genes, we conducted the Gene Ontology (GO) and Kyoto Encyclopedia of Genes and Genomes (KEGG) pathway analyses (Fig. 7B). The KEGG pathway enrichment analysis unveiled that the molecular mechanisms associated with CDC45, and its associated genes primarily revolved around signal pathways involved in the regulation of sarcous function (Fig. 7C). GO enrichment was various aspects The GO enrichment analysis was conducted to investigate the involvement of CDC45 in three aspects of biological processes (BP), cellular components (CC) and molecular functions (MF). The primary focus of enrichment was on contractile fiber, contractile fiber part and muscle system processes, as depicted in Figs. 7D–7F. Furthermore, Gene Set Enrichment Analysis (GSEA) revealed significant enrichment of gene sets related to G protein-coupled receptor ligand binding (NES = −1.808; p adjust = 0.040; FDR = 0.029) and G alpha (i) signaling events (NES = −1.560; p adjust = 0.040; FDR = 0.029), as shown in Figs. 7H and 7I. Altogether, these findings imply that CDC45 plays a crucial role in the regulation of muscle development and function, which may affect the tumor cell invasion and metastasis mechanism of GC.

Figure 7 Functional enrichment analysis of KEGG and GO for CDC45 co-expression genes.

(A) Volcanic plot showed upregulated and downregulated expression level of CDC45 related genes in GC. (B) GO enrichment analysis identified genes involved in GO-CC analysis, GO-BP analysis, GO-MF analysis, and KEGG analysis. (C) Network plot of KEGG and GO enrichment analysis. Interactive network of the top 10 pathways enriched by KEGG pathway enrichment analysis in GC. (D–G) Bubble charts of the top five pathways enriched in (D) cellular component (CC), (E) biological process (BP), and (F) molecular function (MF) via GO pathway and KEGG analysis(G). (H and I) GSEA pathway enrichment analysis.

Discussion

Initially identified as a polypeptide consisting of 650 amino acids in budding yeast (Hopwood & Dalton, 1996), CDC45 has been universally regarded as an indispensable protein necessary for the initiation of DNA replication. The formation of the CDC45-MCM-GINS helicase complex (CMG) holds significance in the early stages of DNA replication in eukaryotes (Hashimoto et al., 2023). CDC45 plays a multifaced role in various malignant tumors. For instance, the suppression of CDC45 demonstrated its anti-tumor ability by impeding cell cycle progression and cell proliferation in hepatocellular carcinoma cells (Feng et al., 2003), while an upregulation of CDC45 was observed in malignant squamous cell carcinomas of the tongue compared to moderate precancerous epithelial dysplasia, with expression levels typically increasing with higher degrees of dysplasia (Li et al., 2008). Although the frequent occurrence of CDC45 overexpression in malignant tumors and its established correlation with poor overall survival has been systematically elucidated in pan-cancers (Lu et al., 2022), further systematic research of CDC45 molecular biological function and clinical application in GC are warranted.

In the present study, a significant increase in both CDC45 mRNA expression level and protein level was observed in GC tissues. Notably, CDC45 demonstrated a robust diagnostic efficacy, displaying high sensitivity and specificity, particularly in the early stages of GC. Intriguingly, patients with lower CDC45 expression exhibited a more unfavorable survival prognosis in GC, consistent with findings in colorectal cancer (Hu et al., 2019), cervical squamous cell carcinoma and endocervical adenocarcinoma (Lu et al., 2022). Taken together, these findings provide compelling evidence supporting the potential of CDC45 as a predictive tumor biomarker for the diagnosis and prognosis of gastric cancer.

CDC45 has been observed to exhibit a positive correlation with the immune cells in KIRC, LIHC, THCA and THYM, while displaying a negatively correlation with immune cells in LUAD, LUSC, and GBM (Lu et al., 2022). However, limited investigations have been conducted on the relationship between CDC45 expression levels and immune cells in GC. Given the significant role of the tumor immune microenvironment in GC tumorigenesis and metastasis, we conducted an analysis to assess the association between CDC45 expression levels and immune infiltration cells, as well as immune checkpoint related genes. Our findings indicated a positive correlation between CDC45 and the abundance of MDSC cells, NK cells, CD4+ cells, mast cells, and macrophages (M0 and M1). Early studies have provided confirmation of the anti-tumor function of NK cells in defending against tumor metastases (Larsen, Gao & Basse, 2014). Activated CD4+ T cells Additionally, recent evidence has demonstrated that activated CD4+ T cells contribute to the maintenance of NK cell viability and their ability to mediate antibody-dependent cell-mediated cytotoxicity (ADCC) via local IL-2 production (Wang et al., 2022). Furthermore, CD4+ T cells have long been recognized for their critical roles in coordinating of innate and antigen-specific immune responses (Speiser et al., 2023). Following polarization from quiescent M0 macrophages, M1 macrophages produce numerous pro-inflammatory cytokines and enhance their cytotoxic effect by increasing the etoposide-induced apoptosis of cancer cells (Genin et al., 2015).

Co-expressed genes typically exhibit similarities in their functions. In order to gain a broader understanding of their overall impact on GC, we conducted an analysis on the correlation between CDC45 and its co-expressed genes. The findings revealed a strong and positive correlation between CDC45 and CDCA5, DTL, NCAPH, RANBP1, ORC1, and RAD54L; additionally, the upregulation of CDCA5, NCAPH and RAD54L was associated with more favorable outcomes in GC patients. To investigate the jointly functional pathways of CDC45 and its partner genes, we performed Go, KEGG, and GSEA enrichment analyses. Result from the KEGG analysis suggested that these genes may enriched in protein digestion and absorption, vascular smooth muscle contraction and neuroactive ligand-receptor interaction. Additionally, the GO analysis demonstrates the involvement of these genes in contractile fiber, regulation of membrane potential and muscle system process pathways. Furthermore, the GSEA analysis reveals enrichment of these gene sets in G protein-coupled receptor ligand binding and G alpha (i) signaling events. It is suggested that dysregulation in these pathways may have a significant association with the progression and development of GC. For example, previous studies have reported the promotion of Nidogen-2in gastric cancer through the protein digestion and absorption pathway (Yu et al., 2019). Altogether, CDC45 and its co-expressed genes involving in the GC malignant metastasis may be attributed to the regulation of smooth muscle cell function, as well as the enhancement of cell adhesion and invasion.

Our study provides a comprehensively elucidation of the underlying role of CDC45 in GC. However, it is important to acknowledge certain limitations. The findings of this study primarily relied on bioinformatics analysis which, although reliable, lack corresponding experimental verification. Firstly, our study solely focuses on the singular gene function of CDC45. Given the multitude of factors involved and regulated in the process of carcinogenesis, intricate interactions among various genes exist, thereby imposing significant limitations on the examination of a solitary gene. Secondly, although we found a significant correlation between CDC45 expression and sensitivity to various chemotherapeutic drugs, the biological mechanism has not been elucidated, which will also be one of the main contents of our follow-up study. Furthermore, although we found that CDC45 is excellent in the early diagnosis of gastric cancer, how it relates to atypical hyperplasia warrants further investigation. Consequently, it is imperative to conduct additional in vivo and in vitro experiments to further investigate and substantiate the reciprocal interactions between CDC45 and its co-expressed genes.

Conclusion

In brief, our study suggests CDC45 upregulates in gastric cancer and provides crucial information for the diagnosis and prognosis for GC patients.

Supplemental Information

Supplemental Information 1 IHC raw data.

Supplemental Information 2 RT-qPCR raw data.

Supplemental Information 3 The data sources in this study.

Supplemental Information 4 The characteristics of GC（n=12）.

Supplemental Information 5 The raw data for co-expressed genes analysis.

Supplemental Information 6 The raw data for GSEA analysis.

Supplemental Information 7 The ROC curves of CEA (A) and AFP(B) in the diagnosis of GC.

Supplemental Information 8 MIQE Checklist.

We are grateful to all the participants of the present study.

Abbreviations and Glossary

CDC45 Cell division cycle protein 45

GC gastric cancer

TCGA The Cancer Genome Atlas

GTEx Genotype-Tissue Expression

GEO Gene Expression Omnibus

RT-qPCR Reverse Transcription Quantitative Polymerase Chain Reaction

HPA Human Protein Atlas

IHC immunohistochemistry

ROC receiver operating characteristic

AUC the area under the ROC curve

GESA Gene Set Enrichment Analysis

EGC early gastric cancer

DEGs differentially expressed genes

GO Gene Ontology

KEGG Kyoto Encyclopedia of Genes and Genomes

FFPE formalin-fixed and paraffin-embedded tissue

MF molecular function

BP biological process

CC cell composition

CDCA5 protein disulfide-isomerase A5

DTL denticle less E3 ubiquitin protein ligase homolog

NCAPH Non-SMC condensing I complex subunit H

RANBP1 RAN binding protein 1

ORC1 Origin recognition complex subunit 1

OS overall survival

IC50 half-maximal inhibitory concentration

ES enrichment score

NES normalized ES

NOM p-val normalized p-value

STAD Stomach adenocarcinoma

KIRC Kidney renal clear cell carcinoma

LIHC Liver hepatocellular carcinoma

THCA Thyroid carcinoma

THYM Thymoma

LUAD Lung adenocarcinoma

LUSC Lung squamous cell carcinoma

GBM Glioblastoma multiforme.

Additional Information and Declarations

Competing Interests

Author Contributions

Human Ethics

Data Availability

The authors declare that they have no competing interests.

Lihua Wu conceived and designed the experiments, performed the experiments, analyzed the data, prepared figures and/or tables, authored or reviewed drafts of the article, and approved the final draft.

Gan Gao conceived and designed the experiments, analyzed the data, prepared figures and/or tables, authored or reviewed drafts of the article, and approved the final draft.

Hui Mi conceived and designed the experiments, analyzed the data, prepared figures and/or tables, and approved the final draft.

Zhou Luo analyzed the data, authored or reviewed drafts of the article, and approved the final draft.

Zheng Wang analyzed the data, authored or reviewed drafts of the article, and approved the final draft.

Yongdong Liu analyzed the data, authored or reviewed drafts of the article, and approved the final draft.

Liangyan Wu analyzed the data, authored or reviewed drafts of the article, and approved the final draft.

Haihua Long analyzed the data, authored or reviewed drafts of the article, and approved the final draft.

Yongqi Shen analyzed the data, authored or reviewed drafts of the article, and approved the final draft.

The following information was supplied relating to ethical approvals (i.e., approving body and any reference numbers):

The Ethics Committee of Affiliated Liutie Central Hospital of Guangxi Medical University approved this study (Ethics Approval Number KY2023-075-01).

The following information was supplied regarding data availability:

The raw data are available in the Supplemental Files.

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
