# Peer review of "Validation of CDC45 as a novel biomarker for diagnosis and prognosis of gastric cancer"

_PeerJ, doi:10.7717/peerj.17130_

## Round 0.1 · original submission · Major Revisions

The reviewer identified a major weakness in the manuscript, which is the insufficient elaboration on the bioinformatic analysis. The author's immunohistochemistry (IHC) results lack specific details, and enhancement is possible through the implementation of a more comprehensive experimental design and explanation.

**Language Note:** PeerJ staff have identified that the English language needs to be improved. When you prepare your next revision, please either (i) have a colleague who is proficient in English and familiar with the subject matter review your manuscript, or (ii) contact a professional editing service to review your manuscript. PeerJ can provide language editing services - you can contact us at copyediting@peerj.com for pricing (be sure to provide your manuscript number and title). – PeerJ Staff

Reviewer 1 ·

Basic reporting

No comments

Experimental design

No comments

Validity of the findings

The conclusion section is not appropriately justified in the manuscript.

Additional comments

Enrichment analysis was not appropriately justified.
The manuscript required significant revision.

Annotated reviews are not available for download in order to protect the identity of reviewers who chose to remain anonymous.

Reviewer 2 ·

Basic reporting

No comments- Please refer to Additional comments.

Experimental design

No comments- Please refer to Additional comments.

Validity of the findings

No comments- Please refer to Additional comments.

Additional comments

Comments to the Author:

This manuscript aims to identify and validate CDC45 as a new biomarker for diagnosing and predicting the prognosis of gastric cancer. The study analyzed the mRNA expression of CDC45 using the TCGA database and the TCGA & GTEx dataset. The protein expression of CDC45 was evaluated using the Human Protein Atlas (HPA) and was subsequently verified through immunohistochemistry (IHC). Additionally, 12 cases of gastric cancer tissues and paired adjacent tissues were used for IHC analysis. To investigate this experiment, the author performed various bioinformatics analyses. In conclusion, the result suggests that CDC45 is upregulated in gastric cancer and is associated with immune cell infiltration, making it a potential prognostic diagnostic biomarker for GC, which could be a better therapeutic approach.

The primary strength of the manuscript is that it is well written and lies in validating CDC45 as a new biomarker for the diagnosis and prognosis of gastric cancer. This approach could be an effective therapeutic strategy for GC. However, the experimental approach could be improved for more informative results.

The major weakness in the manuscript is experimental design, with a lack of detailed information on mechanics. Please refer to the comments below:

Major points:

1. The title of the manuscript is “Identification and Validation of CDC45 as a novel biomarker for diagnosis and prognosis of gastric cancer”. I suggest starting the manuscript title with "Validation" since only CDC45 was evaluated and validated without considering other proteins.
2. An introduction should be more elaborative.
3. At line nos. 54 and 55; Recent studies have…… expressed in various tumors (please provide references)
4. For line no. 76; Provide the downloaded data in Excel and their accession number for clarity of detail.
5. Explain tissue preparation for RNA extraction.
6. Clarify the type of real-time PCR used, including probe sequences if it is TaqMan chemistry and thermal conditions. Primers used in qRT-PCR are in-house designed or taken from other studies. If it has been considered from other publications, then provide the reference number, and in case it is in-house designed, then explain how the primers are designed.
7. Please provide the details (cat no., manufacturer, country) for all chemicals and reagents used (e.g., RNA extraction kit, DNase, etc.)
8. Explain the details about FFPE tissue specimens and how tumor ROIs were analyzed.
9. Please provide detailed information for all the IHC antibodies used and the imager information (version, manufacturer, country). Explain the details of DAPI and the mounting medium. Explain the IHC staining protocol in detail.
10. Lines nos. 112-114 should be under separate headings. Please explain the inclusion and exclusion criteria.
11. At line no. 116: Please mention no. of samples enrolled in the somatic mutation data of TCGA-STAD.
12. Recheck line no 117: GDC query-Maf () function with the pipelines.
13. Rewrite the sentence at line no. 118 and 119; information needs to be included.
14. cBioPortal link needs to explain how the co-expressed genes were identified. Also, explain on what basis co-expressed genes were selected and provide raw data.
15. Explain GEPIA2.
16. What are the samples used for DEGs?
17. Please provide all the raw data in an Excel sheet used for the volcano plot, heatmap, and GSEA.
18. What are the criteria for gene selection used for the analysis in GSEA? Please provide raw data.
19. Please provide the version of the bioinformatic software used for analysis with detailed information.
20. Please mention the no. of subjects considered for the present study from the TCGA database used for the expression of CDC45. Provide the link used for analysis. Explain the difference in Figure 1 (b) and (c).
21. Explain the fold change differential expression of CDC45 through qRT-PCR in Figure 1 (J)
22. Explain how the gene ratio graph has been drawn.
23. Explain in detail how the figure has been retrieved from HPA to distinguish it from normal to GC tissue.
24. In line 166, In the previous studies (provide reference).
25. For Figure 3 (A), explain how the data retrieved for the ROC in the manuscript and provide raw data or its related link.
26. Also explain other gastric tumor markers and then compare with CDC45.
27. At line no. 216; Explain how data has been retrieved to analyze CDC45 expression.
28. Explain about CIBERSORT and ESTIMATE methods.
29. Explain Figure 4 in detail. It seems in Figure 4 (B-E). High sensitivity to the drug is in the low expression group.
30. At line no. 239; Only the data from bladder urothelial has been considered for analysis using the CBioPortal web; is there any explanation for that?
31. Explain other highly abundant differentials and related proteins to CDC45.
32. Please explain all the figures in detail under the legend section for easy understanding.
33. Provide all the abbreviation lists.

---

## Round 0.2 · accepted · Accept

I am satisfied with the modification made and addressing the reviewer's comments. Now, both reviewers are happy with the revised version of the manuscript and hence I consider it for publication.

Reviewer 1 ·

Basic reporting

The manuscript is clearly presented, and all the suggested comments are made.

Experimental design

Every question is addressed properly; it is much appreciated.

All the necessary changes have been made, and a clear and detailed description of the work has been provided.

Validity of the findings

All requested data is presented in the supplement file and clearly answers all the comments.

Additional comments

I will recommend this article for further consideration for publication based on the journal guidelines.

Reviewer 2 ·

Basic reporting

Good

Experimental design

Good

Validity of the findings

Good

Additional comments

The author has provided comprehensive and detailed explanations to the reviewer, addressing all of the queries and concerns raised during the review process. The explanations are likely to be helpful in clarifying any questions or uncertainties that the reviewer had about the content or presentation of the material.

Thanks